# An Adaptable Device for Scalable Electrospinning of Low- and High-Viscosity Solutions

**Ryan J. McCarty** [1,2,*] and **Konstantinos P. Giapis** [2]

1  Department of Chemistry, University California, Irvine, CA 92697, USA
2  Department of Chemical Engineering, California Institute of Technology, Pasadena, CA 91125, USA
*  Correspondence: rmccarty@uci.edu

**Abstract:** This paper summarizes the design and construction of an adaptable electrospinner capable of spinning fluids over a large range of viscosities. The design accommodates needless electrospinning technologies and enables researchers to explore a large range of testing parameters. Modular parts can be exchanged for alternative versions that adapt to the research question at hand. A rotating drum electrode immersed halfway into a solution bath provides the liquid film surface from which electrospinning occurs. We tested and assessed several electrode designs and their electrospinning performance at higher (<500 poise) viscosities. Relative humidity was found to affect the onset of electrospinning of highly viscous solutions. We demonstrate robust device performance at applied voltage up to 90 kV between the electrospinning electrode and the collector. Design and fabrication aspects are discussed in practical terms, with the intent of making this device reproducible in an academic student machine shop.

**Keywords:** electrospinning; viscous solution; nanofabrication; nanofibers; instrumentation; high voltage; practical design; Taylor cone

## 1. Introduction

The application of nanoscale materials for solving an expansive range of problems is limited by the difficulty of scaling up their production [1]. Electrospinning is a technique proven to produce nanoscale materials inexpensively and in large volumes, although for a limited range of materials [2–6]. New techniques to expand the capabilities and capacity of electrospinning increase the potential applications of nanotechnology at relevant scales.

In its most basic description, electrospinning permits the formation of sub-micron diameter fibers from "spinnable" liquids with suitable properties (such as molecular structure, viscosity, surface tension, and conductivity that fall within an acceptable range) using electric fields. Electrostatic forces assist in the formation of a liquid jet, which is further deformed by a whipping motion caused by electrostatic charge imbalances along the jet, resulting in a reduction in jet diameter. If the liquid jet dries or cools sufficiently during its flight from starting point to collection point, its size and morphology can be preserved. Comprehensive information on the theory, current technology, and practice of electrospinning is well summarized online at electrospintech.com [7].

Most research-scale designs feature a hypodermic needle or capillary tube, through which a controllable amount of liquid can be forced out. Electric fields generated at the needle tip are advantageous for initiating a Taylor cone, from which a jet of liquid is ejected [8]. The number of Taylor cones, which can form at a needle tip, limits the capacity [9]. The size of the needle limits the volume, viscosity, and materials that can be pumped to the tip [10]. Some solutions can clog the needle, others can become viscous enough, or dry on the tip, such that Taylor cones cannot be sustained [11]. Relative to electrospinning's use and application, reports on device design are infrequent [12–14].

While capillary designs typically perform well for optimized solution compositions, it is nontrivial to scale-up needle electrospinning for higher yields.

### 1.1. Needle-Free Electrospinning

Needle-free electrospinning offers a solution to many of the drawbacks of capillary designs, enabling larger volumes and a wider range of materials to be produced. A well-known design is the roller electrospinner [15], in which Taylor cones form on the surface of a rotating drum partially submerged in a bath of the material to be spun. A wide range of other concepts, where the spinning solution is placed onto some physical shape, which is advantageous for initiating or sustaining Taylor cone formation, have been proposed [7,15–25]. Typically, these focus on improving a specific feature or introduce a novel design, and quantify its application to a specific polymer of interest. Some large-scale devices have been built for optimized polymer compositions most frequently intended for filtration, biological, or biomedical applications.

A subset of methods includes assisted (or facilitated) electrospinning, in which Taylor cone formation will not self-initiate and must be physically triggered because the static parameters of the system are outside of, but close to the "spinnable" range. Examples include forcing air across the solution surface [19,26], spinning from the surface of introduced bubbles [21,27], or templating Taylor cones by removing objects from a liquid surface [28,29].

### 1.2. An Ideal Electrospinning Device

An ideal electrospinning device for a research setting should be easy and safe to operate, should allow tests with all possible experimental parameters as controllable variables, should produce large volumes of material easily, and should be scalable to industrial production volumes. Furthermore, the device should be easily adapted or improved upon to allow research groups with modest fabrication skills to explore system variables outside of those achievable with commercial systems in a cost-competitive fashion.

### 1.3. Our Device

In this paper, we describe the construction of a device designed to offer researchers a flexible testing environment to explore electrospinning under a wide range of conditions. Although this device does not meet all characteristic of an ideal device, it offers practical solutions to known problems. The device can initiate Taylor cones in both unassisted and mechanically-assisted electrospinning at low (~3 poise) or high viscosities (~500 poise). The design features a rotating drum electrode partially submerged in a liquid bath and a rotating collector drum located directly above. Additional design constraints include: high voltage differential (~90 kV) between electrode and collector, competitive production rates (1–2 g per hour), high-speed rotating collector drum, inexpensive fabrication, and safety features. Particular attention is given to describing designs and materials used, such that academic research groups with access to a machine shop could replicate and improve upon this work.

## 2. Materials and Methods

### 2.1. Design Overview

Our device consists of a frame which supports a hanging collector assembly and positions an adjustable scissors stand, which supports the electrode. Each part is detailed in its sub section. Figure 1 labels the parts of the device. Photographs of the device are provided in Figures A1–A5.

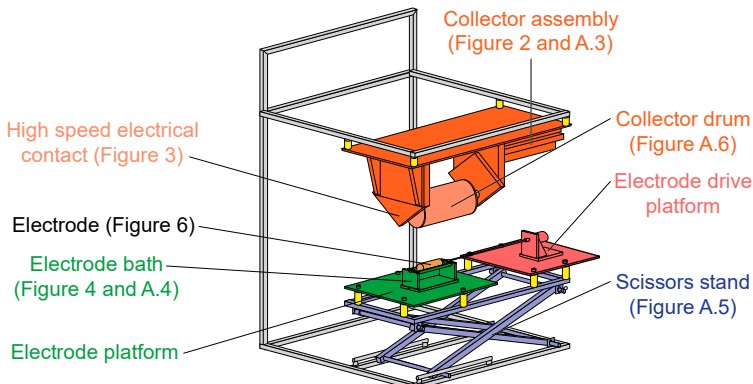

**Figure 1.** A schematic of our electrospinner device detailing parts and associated figures. A collector assembly (orange) hangs from a steel frame and positions a rotating collector drum (light orange). An electrode, from which electrospinning occurs, sits in an electrode bath on the electrode platform (green) which is rotated by a drive motor on the electrode drive platform (pink). A scissors stand (blue) allows the electrode to be raised and lowered relative to the collector drum.

## 2.2. Frame

We built a sturdy "C" shaped metal frame by welding together square steel tubing, using steel rods to make the guiding track and pivot points for the scissors stand. We used a metal miter saw with an aluminum oxide blade, angle grinder, drill press, and MIG (Metal Inert Gas) welder. The top of the frame was originally designed to support various hot air furnaces and temperature monitoring equipment, but in practice, these were not needed. The scissors stand (also known as a scissors jack, scissors lift, stand lift, double scissors lift, lift table, or jack platform) is described in Appendix A.2: Scissors stand description. Holes were drilled through the steel frame to attach glazed ceramic electrical-insulating female threaded standoffs.

## 2.3. Dielectric Materials and Fabrication

Most of the dielectric parts were constructed from King StarBoard® (https://www.kingplastic.com/), a high-density polyethylene plastic intended for marine and boating applications. Starboard® is machinable with woodworking tools, provides suitable strength, and was available at lower prices than materials intended for scientific applications. StarBoard® pieces were assembled using interlocking designs, strategically placed metal screws, and 1/4"-20 nylon socket cap screws. For mounting the high-speed bearings and the electrode platform, PolyOxyMethylene (also known as POM, acetal resin, or most often as Delrin® a product of DuPont™) was used. Polyoxymethylene parts were assembled with 1/4"-20 nylon screws. The majority of this fabrication was done using a table saw, a handheld cornering tool, handheld drill, drill press, and lathe. The cornering tools, sold by Veritas® Tools Inc "Cornering Tool Set" or by other companies as edge rounding tools, are simple cutting tools that remove the sharp edges of the StarBoard® to produce smooth rounded edges without fine burs. Relative to routing, sanding, or filing, the handheld tool was most likely to produce edges that would not initiate corona discharge at high voltages and was also significantly faster.

## 2.4. Rotating Collector Assembly

The collector assembly hangs from the steel frame and houses a drive mechanism and a rotating collector drum (referred to as a mandrel or bobbin in some literature) which we designed to be easily exchanged by the device operator. Additional design constraints imposed by the operators include biasing the top collector to up to ±30 kV, and rotational speeds on the collector drum of 100 Hz (6000 rpm). Collector drums were made from solid aluminum rod (2" diameter version), or aluminum tube welded onto an axel for larger versions (the fabrication of which is described further in Appendix A.3: Rotating collector drum fabrication). A flat plate that hangs from the bearings was

also designed but not used in this study. The rotating drum was held between two R12 ball bearings mounted in polyoxymethylene panels orthogonal to the drum's axis of rotation. The left bearing panel was attached to fixed supports and includes the high-voltage electrical contacts. The right bearing panel is attached to an assembly with horizontal movement towards, or away from the left panel. The assembly includes a **load bearing support** (Figure 2, blue) for the right bearing panel which slides in an enclosed track, a **drive assembly** (Figure 2, yellow) consisting of a polyoxymethylene drive shaft attached to a motor that slides independently in a track parallel to the load bearing support, and push/pull **toggle clamps** (Figure 2, purple) that allow an operator to lock the rotating collector drum in place. To further separate the steel toggle clamps from the high voltage parts, a ceramic spacer was inserted between the polyoxymethylene panel and the toggle clamps push/pull plunger. An additional **stabilizing bearing** (Figure 2, green) was added in the sliding mount along the drive shaft to limit vibrations caused by natural frequency resonance in the somewhat flexible drive shaft. A stabilizing bearing position slightly off the fundamental anti-node of the vibrating shaft reduced the vibrations.

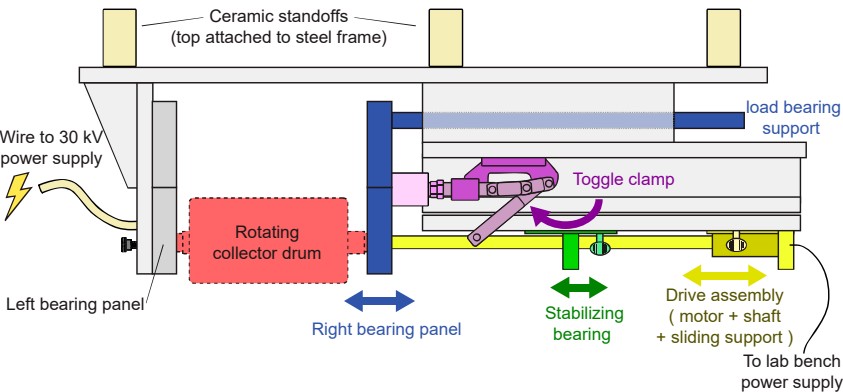

**Figure 2.** Collector assembly, which positions and rotates the collector drum. The collector drum can be easily removed by pulling the toggle clamp and releasing the drive assembly. The text describes the parts in more detail. Figures A1–A3. provide photographs of the assembly.

At high speeds, constant electrical connection to the spinning drum was difficult to achieve with a slip ring design, typically discussed on ResearchGate [30] and employed previously in the laboratory. Instead, we built a contact pin which resembles a dead center on a lathe (Figure 3). In this design, the high voltage wire was soldered to a non-rotating pointed copper rod, which was spring loaded to push into the rotating shaft. In anticipation of the pointed copper rod wearing down over time, a thumb screw (Figure 3, black) allowed adjustments to the spring, in turn adjusting the pressure it applies to the rotating shaft. Minimal wear has been observed so far. Care was also taken to electrically connect the outer races of the bearings to minimize arcing which could damage the bearings.

The majority of the operator's interaction with the device is loading and unloading the sample collector drum, so this step was designed to be easy to accomplish. The collector drum shaft is set into the left bearing and held in place with the operator's left hand. The right hand slides the drive shaft (and entire drive assembly) towards the collector drum so that the attaching bolt can be screwed into position by hand rotating the drive shaft. The collector drum is now temporally supported by the left bearing and drive shaft, so the operator is free to remove their hands. Using the toggle clamp levers, they adjust the right bearing into place. Friction on the toggle clamps is sufficient to hold them in place without additional locks or latches. A thumb screw on the sliding supports for the drive motor can then be tightened, locking the drive assembly in place which helps helped minimize vibrations. These steps are done in reverse order to remove the drum.

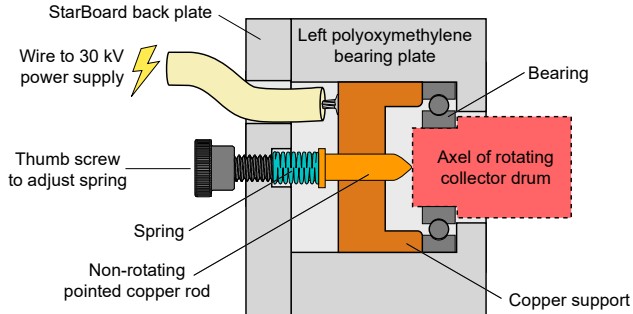

**Figure 3.** A diagram of the high voltage contact to the high-speed rotating collector drum. The high-voltage supply wire is soldered to a copper support which makes contact with the spring-loaded non-rotating pointed copper rod and the outer races of the bearing. The non-rotating pointed copper rod makes contact to the center of the collector drum's axel.

*2.5. Bottom Table, Bath, and Drive Assembly*

We built two platforms on top of the adjustable scissors table, one of which supported the bath and electrode, and a second which supported the lower drive assembly. Using 1/4"-20 nylon bolts we attached polyoxymethylene sheets to glazed ceramic standoffs and the standoffs to the metal frame of the adjustable scissors table. A gap between the platforms ensures electrical isolation of the drive motor (see Figure 1, green and pink platforms). Only a polyoxymethylene drive shaft crossed the platforms. We built a simple support to hold geared DC drive motors (2, 6, 10, or 12 rpm) which were held in place by their weight.

For the electrode platform (Figure 4), we designed a removable bath out of StarBoard®, copper, and PolyVinyl Chloride (PVC) Figure 4. The bath was easy to position or remove, held in place by its weight and a strip of double-sided tape. We constructed the bath with StarBoard® panels as supports fastened together with metal screws. A standard **PVC water pipe** was cut in half lengthwise to produce a trough. A semicircle slot 1/8" deep was lathe cut into the StarBoard® supports, such that the PVC water pipe trough could be rotated into place. The fit of the PVC and the lathe cut groove was tight enough that extra sealing was not needed to make it waterproof, and the PVC could be removed if needed. Electrodes were made with two female 1/4"-20 holes tapped in them, and are discussed in more detail in the discussion section. The StarBoard® has a **through hole bracket** for a 1/4"-20 **nylon screw**, which supports the left side of the electrode. We built a **contact pin** for the electrode from a 1/4" copper rod, which was threaded on one side and left straight on the other. The contact pin was screwed into the right side of the electrode, attached to the drive shaft via a coupling on the left side with a set screw, and supported in the middle by the supportive slip ring (see Figure 4). The **supportive slip ring** was made from a cylinder of copper with a 1/4" through hole, and a long 1/8" rod attached to its bottom. A highly practical part of the slip ring support is that it can be inserted or removed vertically from its housing, which allows the electrode to be easily removed so long as the left side supportive screw was unscrewed and the coupling was loose.

To supply the high voltage to the electrode, an interior 11/64" hole was drilled through the base of the removable bath, and through the right support. We soldered together two 5/32" × 0.014" copper tubes (manufactured by http://www.ksmetals.com/) and inserted this into the base and right support before assembly. The long 1/8" rod on the supportive slip ring and a 1/8" rod on the high-voltage line could then "plug" into the 5/32" tubes. A very slight bend to the 1/8" male rod allows a firm fit in the 5/32" female tube which is easy to remove by hand but would not slide out on its own.

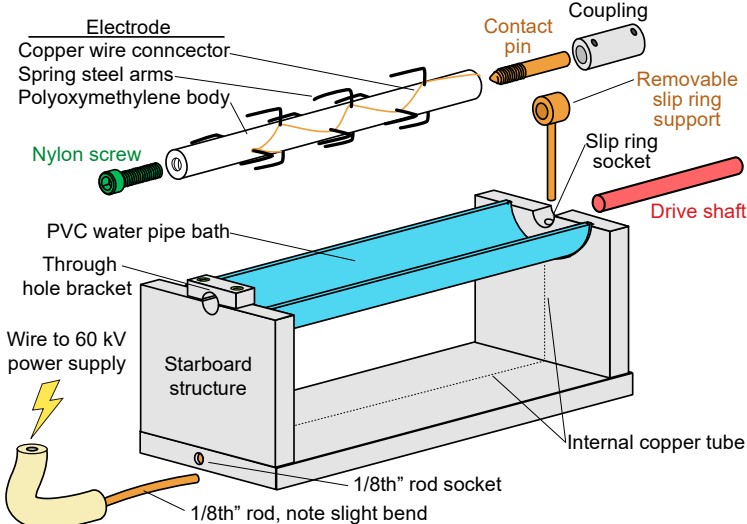

**Figure 4.** A semi-exploded view of an electrode and the electrode bath. A photograph of this part is pictured in Figure A4. To assemble, the **contact pin** is screwed into the **electrode** and pushed into the **removable slip ring support**. The removable slip ring support is pushed into its socket, which guides the electrode and contact pin into the PVC water pipe bath. The **nylon screw** is passed through the **through hole bracket** and screwed into the electrode. The **coupling** can then attach the contact pin and **driveshaft** together. The **1/8″ rod** is then plugged in.

## 2.6. Electrodes

We tested many different types of electrodes for their capacity to spin viscous liquids. Copper wire, copper tubes, monolithic aluminum, polyoxymethylene, and spring steel wires were used to design a variety of previously described and new electrodes. The designs for electrodes are overviewed and discussed in the results section.

## 2.7. High-Voltage Wires

Wiring from the high-voltage power supplies to the device was given extra consideration. Most of the wire was high-voltage 60 kV DC wire, composed of copper strands coated in an uncharacterized metal encased in silicone. For the higher voltage line, 60 kV DC wire was enclosed in a polymer sheath, typically PolyTetraFluoroEthylene (PTFE). Interconnects were directly soldered together, or connected with homebuilt connectors made using copper rod threaded or tapped with 1/4″-20 (detailed in Figure A7). This required rotating the wire to "plug" it in, but ensured that joints along the high-voltage transmission lines would not accidentally become unplugged. Joints were sealed with DuPont™ Kapton® tape interior with standard electrical tape exterior, and insulated with a polymer tube (typically PTFE) outer shell for extra precaution.

All of the soldering was done using standard soldering irons, even the larger copper pieces which took a while to heat to soldering temperatures. The convenience of soldering to the copper and its high electrical conductivity far outweighed the material cost and extra patience needed to machine copper.

## 2.8. Power Supplies and Environmental Controls

We incorporated several commercial devices into our design. A low voltage variable DC power supply was used to control the high-speed drive motor. The bottom electrode drive motor was powered by an AC to DC outlet plug power adapter, or a variable lab bench power supply. For high-voltage power supplies, we used a Spellman SL2000 (https//spellmanhv.com/) that was capable of reaching +63 kV and a Glassman High Voltage Inc. "series ER" that was capable of reaching −30 kV. These 2000s-era devices contained several safety features, such as easy to interface safety interlocks, multiple

switches to turn on high voltages, and an automatic shutoff after arc detection (on the Spellman device), which we found useful on several occasions.

During electrospinning tests, we realized the importance of relative humidity in the spinning environment. Some viscous solutions were particularly sensitive to relative humidity, refusing to initiate Taylor cones at high humidities, but producing numerous fibers at low humidities. We had the highest success when spinning on windy, dry Southern California days (relative humidity <30%), and typically monitored weather to plan our tests and experiments for optimal conditions. However, during periods of higher humidity, a portable dehumidifier was added into the fume hood, and occasionally space heaters or heat guns were used to control relative humidity by raising the air temperature.

### 2.9. Testing and Measurement

Standard handheld FLUKE® voltmeters with a high-voltage probe were used to take readings and to test high voltages. The remaining tests were taken from a distance. On each of our collector drums, we used black permanent markers to color a dark section so that a handheld speed reader could be used to measure rotational frequency. A handheld temperature gun was used for measuring temperatures of the solution and other parts of the device. Inexpensive battery-operated humidity and temperature monitors with large displays were also placed throughout the fume hood, and the readings were taken from a distance. We used a Canon EOSRebelT5i with a 75–300 mm zoom lens on a tripod to investigate corona leaks in the system and characterize electrospinning performance from a safe distance.

For electrospinning performance testing, a variety of polymer solutions were used, most frequently a polymer: polyvinyl alcohol (known as poly(1-hydroxyethylene) or PVA), polyvinylpyrrolidone (known as 1-ethenylpyrrolidin-2-one or PVP), and/or $CsPO_3$ (cesium metaphosphate), was hydrated with water, methanol, or a mixture of water and ethanol, and $CsH_2PO_4$ (cesium dihydrogen phosphate) was added to adjust conductivity. To synthesize our solutions, we used graduated cylinders, syringes, laboratory scales, hot plates, magnetic stirrers, and hand-mixing or shaking. We quantified our solutions with a handheld solution conductivity meter, and an AR-G2 rheometer produced by TA Instruments Ltd. Reported viscosity values are dynamic viscosity. Cesium metaphosphate solutions are advantageous for infrequent testing, as solutions are easy to rehydrate with warm water, easy to remove from parts when dry, and do not facilitate mold growth. A supplemental video is available to demonstrate how a vicious solution of cesium metaphosphate and water can be created from cesium metaphosphate and water precursors. We used microscopy to characterize the spun fibers, employing a tabletop optical microscope and a ZEISS 1550VP FESEM.

## 3. Results and Discussion

### 3.1. Results and Discussion Overview

We used our device to electrospin and collect a variety of μm and nm sized fibers from solutions with low and high viscosities. When fibers were produced, we typically observed long straight fibers with sizes between 35 and 1200 nm, occasionally with morphologies as large as 34 μm and as small as 25 nm. The produced fibers were strongly dependent on solution parameters (such as conductivity, compositions, surface tension, etc.), and environmental parameters (such as relative humidity). Demonstrations of the device on specific compositions and parameters are reported in Appendix B including microscopy images of synthesized fibers, statistics on fiber diameters, and details on the compositions used to produce them.

The focus of this paper is the design of the device. In the following sections, we overview the design performance and discuss the qualities of tested electrodes (rather than application of this device to a specific compositions). First, we start with electrical performance, including: high voltages, high-voltage safety, and electric field. Second, we discuss mechanical considerations: the rotating drums, user interface, the flexibility of the system to explore many scientific questions, and related

considerations. We then overview electrode designs and their performance at increasing viscosity. Several sections direct the reader to additional details included in the appendix. The appendix also comments on ideas that did not work in A.7: Unsuccessful design concepts, as well as a statement on costs incurred during fabrication in A.8: Material and time costs. A listing of system parameters and values achieved with this device are listed in Appendix B.3: System parameters.

### 3.2. Device Adaptability and User Interface

The design of our device was adaptable, allowing the operator to explore a large range of experimental parameters. When attempting to electrospin new materials, or to scale materials demonstrated on needle spinners to needless designs, having an expansive range of parameters to explore made the testing and exploration more practical. Most of the advantages originated from (1) the safe enclosed high voltage testing space, (2) the simple flat adjustable height table to hold the bottom electrode, and (3) the sturdy mount for rotating sample collectors.

The interior of the fume hood was larger than twice the size of the electrospinning device, which allowed room for a variety of power supplies, air heaters, or dehumidifiers. The wide range of voltages was useful when working with new solutions to quickly determining if spinning was sensitive to the electric field strength.

The design and procedure described for loading and unloading the sample drum worked well in that it was easy and fast to perform. In practice, the toggle clamps needed to be adjusted so that the right bearing applied a small amount of horizontal pressure, which ensured that the electrical contact pin was making contact with the end of the collector drum. In future designs, creating a pin with larger spring distance would have eliminated this design flaw. If the operator did not want to touch the newly formed fibers, some dexterity was required to unmount and remove the drum by only touching its axel. A collector drum grabbing tool could be designed to make the drum removal step easier.

### 3.3. High-Voltage Performance

The device can be operated at high voltages for long durations of time, a result only achieved after many tests and adjustments. Using the zoom camera lenses with all laboratory lights turned off, long exposure photographs were taken to observe minor and transient corona discharge, which could not be observed with the naked eye. We found this particularly helpful and practical for identifying and addressing corona leaks. Some example photographs of this are included in Appendix A.5: Troubleshooting corona discharge. We did not observe corona discharge on the exterior of the high-speed contact or related parts of the top rotating drum (although the interior high-speed electrical contact point could not be observed). The bottom bath would frequently be a place for corona discharge, having rounded edges on all of the bath parts helped avoid this, but minor nicks or scratches added during routine use would frequently become new locations for discharge. Even in ideal conditions, after operating for some time, stray fibers that landed on the bath or table would occasionally serve as minor discharge points, as would parts of the rotating electrode if solutions dried. These discharge points can initiate arcing across the bottom and top electrodes (and a larger voltage drop or smaller electrode-collector distances could only be reached if we corrected for these discharges).

### 3.4. High-Voltage Safety

Due to the electrical hazards of the high voltages our system could reach, we took several precautions. During device operation, it was not uncommon to initiate arcing across a 12 cm distance between our electrode and collector with more than 60 kV total voltage drop. The electrospinning device was only operated in a dedicated fume hood. Extensive grounding work was performed to ensure that the fume hood and associated parts were well grounded. Both of the drive motor housings needed additional grounded connections along their exterior. For the fume hood, we added redundant spring-loaded contacts on the sliding doors, grounding straps, copper plates, and conductive bolts to provide a highly conductive connection to all parts of the fume hood and the building ground. We

fitted each of the sliding fume hood doors with an open/closed safety switch that was wired into the power supply to ensure that the device could not be powered on if the doors were open. In addition to the legally required safety signs, we created additional signs to help ourselves remember the present danger and to explain the danger to other scientists in the laboratory. Several metallic parts close to the rotating drum were left ungrounded and were known to charge up to 10 kV during peak voltage operation. We used the following safety steps when opening the fume hood:

1.  De-energize the high voltage power supplies.
2.  Set a 5-min timer to allow the device time to discharge before opening the fume hood.
3.  After the timer rings, use a high-voltage grounding rod (also sold under names such as discharge stick or discharge rod) with a high-voltage insulating handle to open the fume hood doors. Then contact each part known to charge with the grounding rod tip, to ensure they had fully discharged.

Device training for operators included standard operating procedure overview, identification of circuit breakers relevant to the power supplies and fume hood, and device fire rehearsal. In the event of a device fire (which occurred once after arcing when working with a solution containing a high percentage of ethanol), absolutely no changes were made to the opening protocol. The potential dangers of the electrical hazards to the operator were determined to be far greater than a fire contained within a fume hood. We believe that rehearsing this scenario would help the operator remember electrical hazards and make safe decisions when responding to a visually alarming fire.

In hindsight, we realize that a dedicated holder to store high-voltage contacts when not properly plugged in would have been useful for establishing habits that make accidentally energizing the system less dangerous.

### 3.5. Electric Field for Spinning

The generated electric fields were acceptable for electrospinning. We observed self-initiated spinning from a cylindrical electrode of a polyvinyl alcohol polymer-water solution at voltages as low as 1.5 kV if humidity and solution characteristic were optimal. During operation, the majority of spun fibers were drawn towards and collected on the collector drum. There was a slight preference for escaping fibers to exit to the left side, but we are unsure if this was primarily due to the different electric field in this direction or air currents within the fume hood. Future designs would likely benefit from incorporating electric field modeling (such as the calculations by Jentzsch et al. [31]) into the design and optimization of the device.

### 3.6. Mechanical Performance of Collector and Electrode

We experienced good results from the motor-driven collector drum and bottom electrode. When testing the high-speed collector drum, several different drums designs were made with diameters from 1 to 6 inches (2.54 to 15.24 cm). With the 6-inch drum, we achieved rotational frequencies of 105 Hz, resulting in a surface velocity of 50 m/s. SEM microscopy showed that fibers produced at these conditions were long and smooth. We believe this indicates that sufficiently electrical connection was established by the spring-loaded pin to the collector drum, although it is likely conductivity through the bearing was also assisting this connection. Some rotational frequencies were near to the natural frequencies of the device, which would result in significant shaking. We avoided these frequencies, and we believe that better balancing our collectors would have reduced vibrations. We tried several approaches to connecting the drive shaft to the collector drum; the most successful method used a 1/4"-20 steel bolt and taped female holes in the collector drum shaft and the polyoxymethylene drive shaft. A slot cut on both ends of the bolt ensured that it could be unscrewed with a flat head screwdriver if it became stuck.

Single speed DC motors with gearboxes (2–12 rpm, 0.03 to 0.2 Hz) drove the bottom rotating electrode. On rare occasions issues arose when solutions dried out or solidified, causing the rotating electrode to seize. In one case this resulted in the entire bath assembly rotating onto its side, which

could have caused significant problems if the operator had not been present to power down the system. To prevent this, a neodymium magnet coupling was designed, such that if the electrode seized, the extra force would cause the magnets to detach and "break" the drive shaft.

### 3.7. Other Design Considerations

The solution bath design would benefit from something to mitigate solution spills as well as temperature controls. Spills while pouring solutions into the bath and or overfilling the bath would result in some solution on the electrode's platform. In some settings, this would result in a conducting pathway close to or near the metal frame, which could result arcing from the spill to ground. A modified design that incorporates a liquids trap, such as holes or a grove on the platform, would be useful in preventing long conductive pathways. Temperature control of the solution bath was frequently a desired feature, such as described in other designs [32]. For continuous large scale production, mechanisms which maintain the hydration and viscosity of the solution will need to be developed.

### 3.8. Electrode Design and Performance

Several different electrode geometries were designed and tested for a wide range of solution viscosities. Using an optimized low viscosity polyvinyl alcohol spinning solution, we demonstrated electrospinning from every electrode surface (even non-conductive surfaces) that we designed. However, as solution viscosity increased, the capacity and capability of all designs decreased. Conditions for electrospinning viscous materials are typically more limited than non-viscous materials. Electrode designs tailored to specific conditions can expand the possible viscosities which can be spun. For higher viscosities, assisted-electrospinning electrode designs appear to be necessary for higher yields.

We discuss our designs in terms of unassisted-electrospinning and assisted-electrospinning. In unassisted-electrospinning, the rotation of the electrode in the bath primarily functions as a means to coat the electrode with the solution. The rotation is slow, and it is not considered to make an additional contribution to the formation of Taylor cones. In contrast, assisted-electrospinning is used to describe a situation where Taylor cones form only after the solution is mechanically perturbed. In this regard, we primarily investigated the formation of a Taylor cone "templates" by stretching the viscous material into a thin strand (see Figure 5). As the strand thins, it influences the electric field around it and instabilities along the strand cause it to vibrate back and forth. Eventually, the stretched strand breaks and Taylor cones are formed on the broken ends of the strand. Electrospinning will continue from the formed Taylor cones for a limited amount of time until no more solution is available to sustain the Taylor cone, an event which happens faster at higher viscosities.

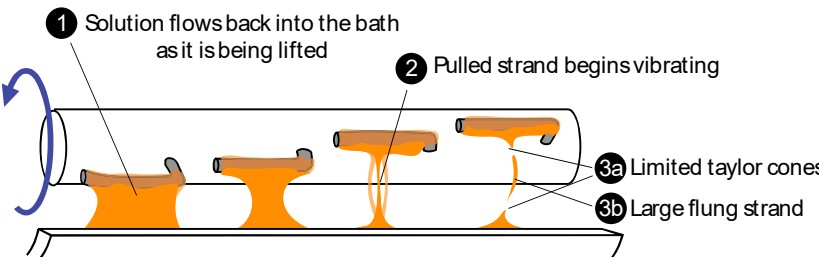

**Figure 5.** A diagram illustrating the assisted formation of Taylor cones from a saguaro electrode. The solution is indicated in orange. The saguaro electrode is detailed later and appears in Figure 4, the bottom of Figure 6, and in Figure A4. (1) As the arms are removed from the bath, the majority of the solution will flow from the arm. (2) The flowing fluid thins and is stretched into a strand, as they are further pulled, they begin to vibrate due to imbalances in the electric field. (3a) After the strand breaks, Taylor cones are formed on the broken ends, and electrospinning jets are observed. (3b) Occasionally, as stretched strand would break, the electric fields would eject large pieces or globs them towards the collector.

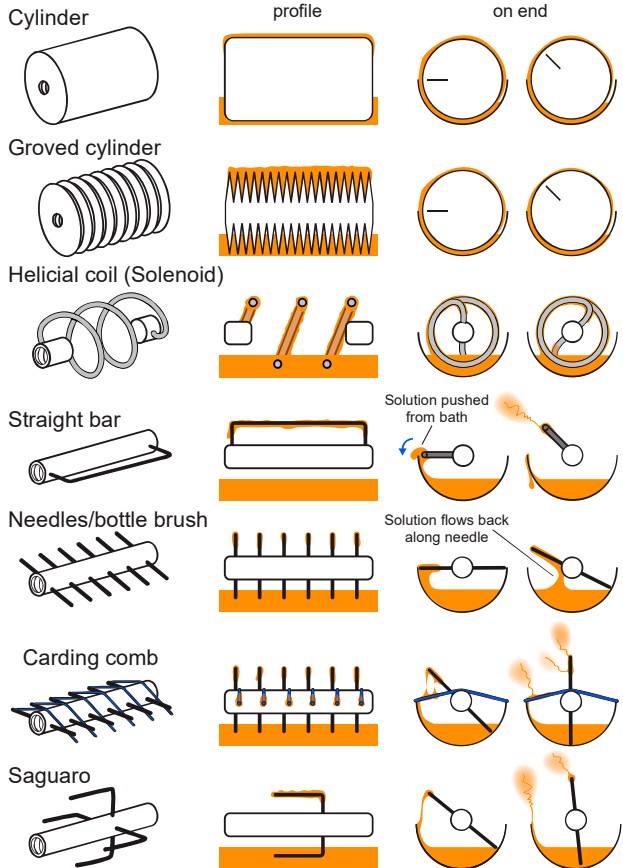

**Figure 6.** Electrode designs characterized by viscous solutions. The bottom designs were the most successful for spinning viscous solutions. Each design is discussed further in the text.

We use the term stronger (or increased) electric field to indicate that either the voltage drop from the electrode to the collector is larger and or that the distance from the electrode to the collector was decreased. We used both methods to explore a variety of conditions, but we did not quantitively model or characterize the electric fields.

*3.9. Unassisted-Electrospinning Electrode Performance*

For unassisted-electrospinning designs, where Taylor cone formation needed to self-initiate, larger monolithic metal electrode morphologies were the first to have difficulties spinning as viscosities increased. **Cylinders** (see Figure 6, Cylinder) and cylindrical drum variants such as the grooved cylinder, were the first to exhibit poor performance as viscosities increased. With increasing viscosity, spinning transitioned from initiating across the whole surface to preferring to spin only from the rounded right and left edges of the cylinder and eventually refusing to spin. This behavior indicates that the edges had advantageous electric fields for spinning relative to the center of the cylinder. Increasing the electric field in some cases increased the viscosities that could be spun (with all other known variables kept constant). However, at stronger electric fields, corona discharge plasmas would form and eject material, typically initiating Taylor cones and electrospinning in the process. It was difficult to tell when unassisted Taylor cone formation had ceased, and the system now relied on a corona discharge event. Higher viscosities (80 poise) or higher solution conductivities suppressed corona discharge relative to the dry setting (same setup without a spinning solution). We suspect that this is most likely the result of the solution coating and smoothing out small nicks or dents where corona discharge would initiate in the dry. It was not uncommon to see what appeared to us as "bubbling" on the surface at the highest voltages, most likely the result of Taylor cone formation and

collapse due to the limited flow of the viscous materials [33]. At further increased electric fields (~93 kV voltage drop with 11 cm electrode collector gap), dielectric breakdown of the air would occur, resulting in intermittent arcing. Designs with surfaces created to intentionally concentrate the local electric field into optimal electrospinning conditions, such as the **grooved cylinder** (Figure 6), did not perform well at high viscosities, as the viscous materials would coat the grooves creating a new surface and disadvantageous electric field. Viscosity, surface tension effects, and solvent drying rates (which would result in changing viscosities, temperatures and surface tensions at the solution surface) complicated attempts to spin from grooved electrode designs.

A small disk (essentially a cylinder with a very narrow length) performed qualitatively equivalent to the longer length cylinders (where spinning happened only on the outer edges).

Unassisted-electrospinning wire designs that did not feature a terminating wire tip performed mildly better at spinning than monolithic designs, however at higher viscosities similar problems to those discussed for cylinders where encountered. Thick and thin gauge copper wires were used, typically terminating in a center axel rod made of polyoxymethylene, steel or copper. Examples include the **helical coil** and the **straight bar** (Figure 6). Small diameter coils with fewer turns per length performed best, as did the right and left edges of all designs. A single loop coil had similar performance to multiloop coils. All designs with horizontal wires, such as the straight bar and saguaro design (discussed below), tended to "push" solution up and over the edge of the bath.

Unassisted-electrospinning wire designs which featured terminating wire tips, such as the comb or bottle brush designs, performed better than non-terminating designs in terms of viscosities that could be electrospun. Typically, spring steel rods or wires were pushed through drilled holes in a polyoxymethylene rod with a thin copper wire providing electrical connectivity. These designs are a close analog to single needle/capillary designs where an advantageously concentrated electric field is created at the needle tip. Due to the concentrated electric field, these designs were a bit more temperamental to work with. The voltage range at which spinning would occur was often close to voltages needed for corona discharge and subsequent arcing to occur. Even with more conservative voltage settings, it was not uncommon for the dynamic nature of viscous solutions to suddenly result in a concentrated electric field at one of the needles, causing arcing to occur. Similar to the grooved cylinder, designs with too many needles would result in a saturated electrode with solution filling in between the needle gaps, generating non-advantageous electric fields. Further improvements, such as optimization of the electric field, or incorporation of micro bubbles as recently demonstrated [21], think it possible that optimization improvements on unassisted designs or incorporating.

*3.10. Assisted Electrode Performance*

At higher viscosities, we found that assisted Taylor cone formation makes it much easier to spin fibers, relative to unassisted-electrospinning designs. We observed this in unintentional settings, such as drops dripping from helical or straight bar designs, as well as an intentional feature as described in Figure 5 for the saguaro design. It is likely that other assisted Taylor cone formation examples found in the literature also perform well at high viscosities [28].

The carding comb design (Figure 6), functioning much like the carding processes from textile fiber production, incorporated a rotating comb or bottle brush design which would pass by an independent set of needles or objects. As a test of this concept, we used a wire cord wrapped through the electrode to create an independent object for the needles to pass by. In the carding-comb design, strands are created and stretched between the non-moving cord and the rotating needles, which eventually results in Taylor cone formation, as described in Figure 5. The rotating needles needed to pass very close to the wrapped cord, or material flows along the electrode surfaces instead of breaking into a thin fiber (as depicted in Figure 6, Needles/bottle brush). When properly adjusted, this design could produce fibers, but its yield was limited relative to the saguaro design.

The saguaro design (Figure 6), which resembles a saguaro cactus, was the most successful design for high-viscosity solutions. As the electrode rotates, it creates strands (Figure 5), often creating several

strands from each arm, eventually resulting in Taylor cones. Although Taylor cones formed on the rim of the bath as pictured in Figure 5, these Taylor cones frequently collapsed, most likely due to insufficient local electric field. With this design, the most common problem was solvent evaporation from the bath. Evaporation would first result in increased viscosity and a minimal decrease in nanofiber yield. Further evaporation would result in solidified material forming on the arms and the edges of the bath, which typically resulted in the electrode binding in the bath.

The saguaro design was less temperamental than the other designs, presenting a robust spinning environment which could generate fibers from every solution we tested. The addition of nm and um sized solid particles to the spinning solution did not decrease the yield of fibers. As solution viscosity approached 500 poise, fiber yield drastically decreased. As viscosity increased, so did the likelihood of flinging large (> 50 um) strands or globs of solution. Due to the dangers of flinging thick conductive pathways, we only tested a few solutions with viscosities over 200 poise. It is possible unintended effects, such as corona discharge at the termination pulled strands, may have caused local heating of the solution, resulting in lower viscosities which enabled spinning. A better-designed electrode should be able to minimize the risk of flinging material and may enable new work on challenging to study viscosities.

We designed additional electrodes, but due to time limitations, we were unable to construct and test them. We describe some of these designs in Appendix A.6: Additional electrode design.

## 4. Conclusions

This work has several implications for the design of electrospinning devices intended for nanofabrication. Our needless device provides a pathway for fabricating larger laboratory scale volumes of nanomaterials from an expanded viscosity range via electrospinning. It also provides a synthesis route for materials likely to clog capillary needle electrospinning devices, particularly solutions containing solid particles. Solutions with particles often require frequent stirring to keep particles suspended and may present challenging viscosities, both issues can be addressed by presented rotating high-viscosity electrode designs. We believe production yields from our design are likely to increase significantly with better-optimized solutions and electrodes. Furthermore, the device was built to test scalable designs in a small environment, and it is likely that nanomaterials designed from similar devices have an increased potential to scale to large production volumes.

As a research tool, devices like ours allow research teams to easily run experiments across a large number of experimental parameters, including electrode designs, collector speeds and designs, electric fields, relative humidities, and air temperatures. Flexibility in experimental devices is particularly useful when testing new materials that have limited demonstrations on capillary needle designs, or even solutions that were impractical to test in such settings. Finally, practical aspects and unsuccessful designs are not frequently reported in the scientific literature. However, because this content is vital to efficiently designing safe, useful devices, we anticipate that our discussion one these topics will be useful for the design of next-generation electrospinning devices.

**Supplementary Materials:** The following are available online at https://zenodo.org/record/3364133#.XUzHCUG-mUl, Video S1: Creating a Cesium Metaphosphate Water solution.

**Author Contributions:** Conceptualization, R.J.M. and K.P.G.; methodology, R.J.M.; validation, R.J.M.; formal analysis, R.J.M.; investigation, R.J.M.; resources, R.J.M. and K.P.G.; data curation, R.J.M.; writing—original draft preparation, R.J.M.; writing—review and editing, R.J.M. and K.P.G.; visualization, R.J.M.; project administration, K.P.G.; funding acquisition, K.P.G.

**Funding:** This work was primarily funded by U.S. Department of Energy, Advanced Research Projects Agency-Energy (ARPA-E), Award No. DEAR0000495.

**Acknowledgments:** We are thankful for the workshop access and assistance provided by Daniel McCarty, chemical reagents provided by SAFCell Inc. Pasadena, and rheometer access provided by Ali Mohraz's lab at UC Irvine. We are also thankful for the thoughtful feedback and comments of two anonymous reviewers.

**Conflicts of Interest:** The authors declare no conflict of interest.

## Appendix A.

*Appendix A.1. Photographs of the Device*

Photographs of the full device are featured in Figure A1. and Figure A2. An additional angle of the right side of the collector assembly is pictured in Figure A3. The electrode, bath, and electrode drive motor are pictured in Figure A4. The following section and Figure A5. detail the parts of the scissors stand.

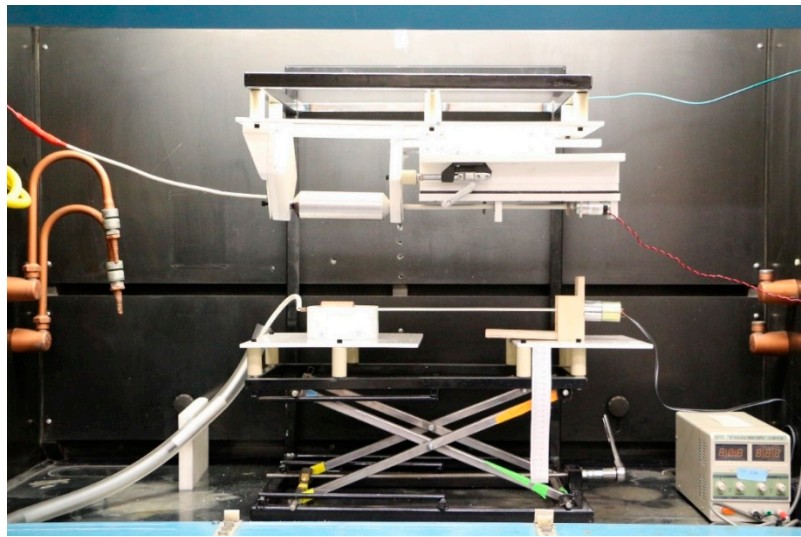

**Figure A1.** Photograph of the entire device. Note that the pictured bath and bottom electrode drive mechanism is an earlier version, later replaced with the improved version featured in Figure A4.

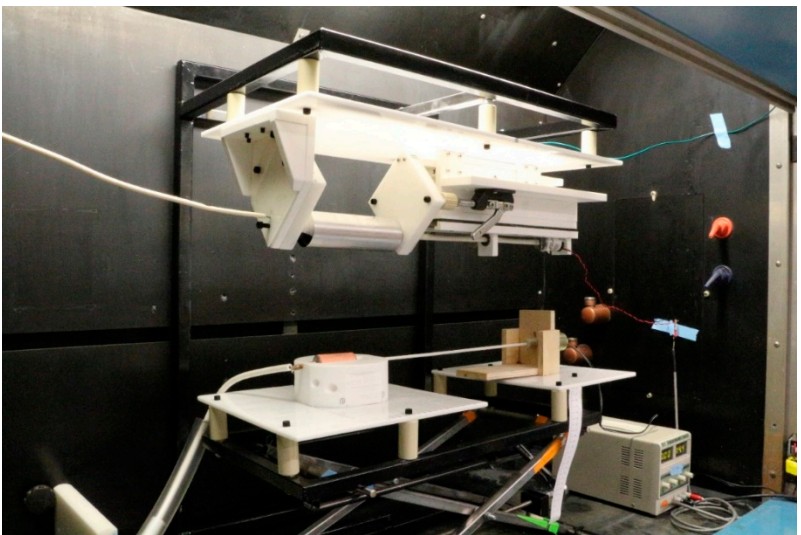

**Figure A2.** Photograph of the device. Note that the pictured bath and bottom electrode drive mechanism is an earlier version, later replaced with the improved version featured in Figure A4.

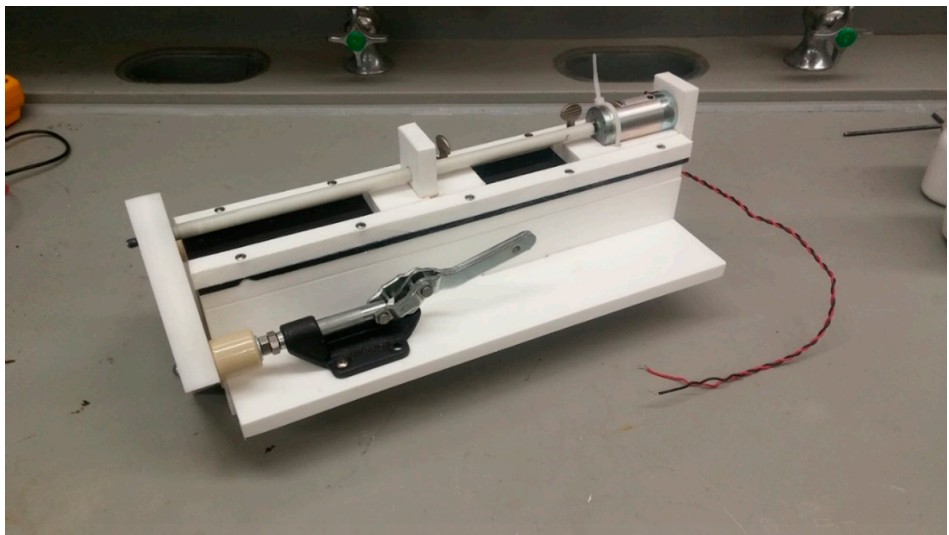

**Figure A3.** Photograph of the right bearing panel, toggle clamps, stabilizing bearing, and collector drive assembly. The assembly is upside down (rotated 180 degrees along its axis) from how it is installed on the device. Refer to Figure 4 for labels.

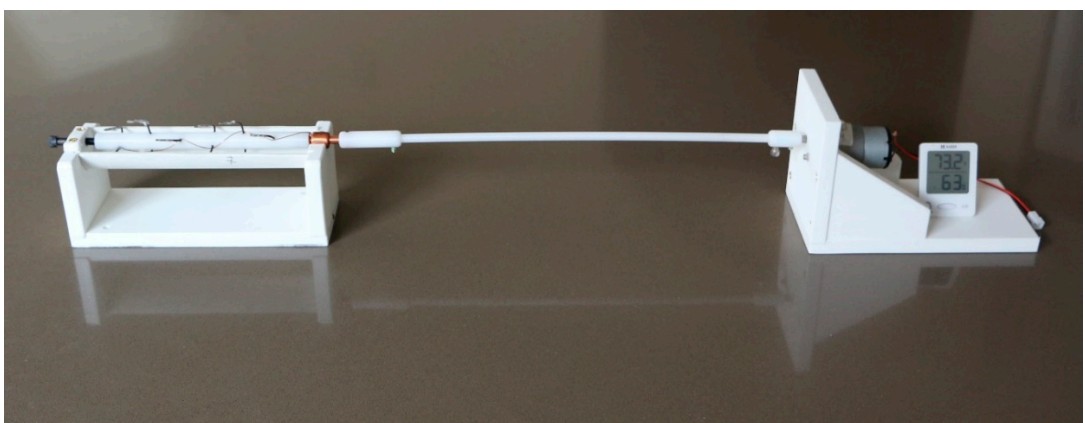

**Figure A4.** Photograph of the bath, a saguaro electrode in the bath and the electrode drive moto. The pictured assembly is approximately 80 cm long (32 inches). A humidity meter is sitting on the drive motor stand.

*Appendix A.2. Scissors Stand Description*

The scissors stand was constructed from two "scissors" made from square tubes with a bolt connecting them. Steel rods joined the scissors, with a spacer tube keeping the pair from coming closer, and cotter pins keeping the stand from coming apart. The right side of the scissors stand was fixed to the metal frame (bottom) and table (top), the left side on top and bottom could slide from right to left in a track. A long threaded stainless-steel bolt attached to a brass block/nut and a fixed nut and socket wrench allowed the height of the table to be adjusted by slowly opening or closing the scissors stand. The brass block/nut was a tapped brass block mounted in a bar connecting the legs of the scissors stand opposite to the fixed nut. Brass was used to minimize the chance of the bolt binding. Some caution should be used prior to attaching the threaded bolt, as scissor stands can indeed function similar to regular scissors, and although parts used here are not "sharp" their large size and mass can crush or shear fingers. A labeled photograph of the scissors stand is presented in Figure A5.

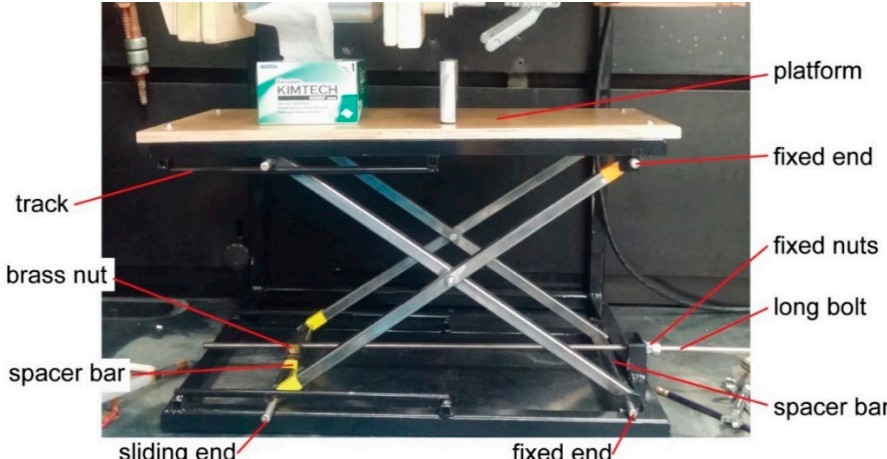

**Figure A5.** The scissors stand we constructed for our electrospinner as assembled on the steel frame. Note that the photograph is not of the final version, and so the long bolt does not yet have a socket wrench attached, cotter pins are not pictured, and the pictured wooden platform was not used in the final electrospinner design. The orange and yellow objects in the picture are unimportant pieces of tape wrapped around the legs or spacer bar. Scale can be interpreted from the box of Kimtech kimwipes or the pencil (right side bottom).

*Appendix A.3. Rotating Collector Drum Fabrication*

We found the collector drums to be the most challenging parts to fabricate, due to their size, shape, and rotationally balanced requirement. Spinning at slower speeds would lessen the demands on this part. Figure A6 illustrates the discussed parts. We started with an aluminum axle rod and lathe-turned the ends for our bearings. We also cut a step to help position the disks. Using large 1/2" thick aluminum disks, we drilled a center hole such that these could slide onto the axel. The disks were TIG welded onto axel to produce the "Barbell". The barbell was then lathe turned, machining the aluminum disks until the extruded tube could slide over them. We then cut a piece of extruded aluminum tube, slid it over the disks, and TIG welded the tube onto the disks. This welding was challenging due to the thin wall of the tube, large mass of the disk, and curved surface. The assembled collector was then lathe turned and balanced.

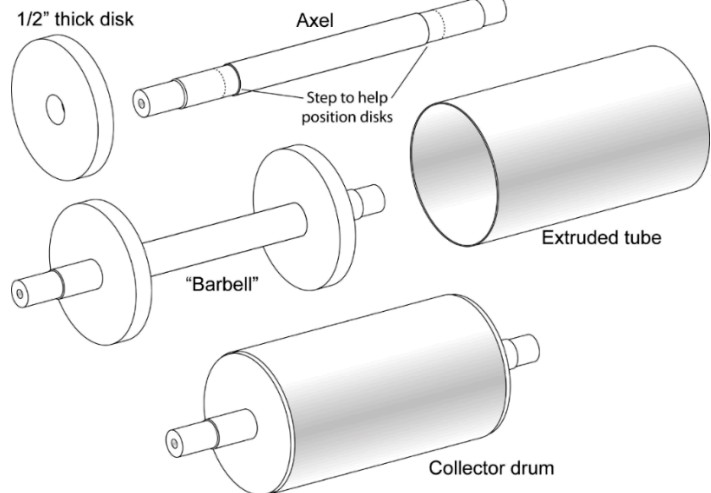

**Figure A6.** Schematic of the collector drum being constructed. Two $\frac{1}{2}$" thick disks are placed on either side of the axel. An extruded tube is then slid over the disks. All parts were TIG-welded together.

*Appendix A.4. High-Voltage Connector*

A schematic of our high voltage electrical connector is presented in Figure A7. Typically, the thick PolyTetraFluoroEthylene (PTFE) sleeve was ~25 cm in length (~10 inches). When the threaded copper was fully screwed in, the electrical tape on the wire formed a tight seal against the thick PTFE. This seal may have helped limit surface creep from inside the connector to the exterior of the wire.

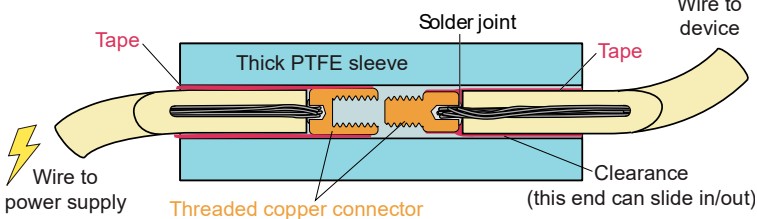

**Figure A7.** Cross-sections of a high voltage electrical connector. The thick PTFE sleeve was typically 2.5 times longer than pictured.

*Appendix A.5. Troubleshooting Corona Discharge*

Figure A8 is a 5-s exposure photograph of a later-replaced high-voltage contact design. The locations of small corona discharge events can be identified in the photograph, which can be used to identify and correct issues on the part. Figure A9 is dotted with purple light from corona discharge events. These events originate from the sharp tips of electrospun fibers which have been sprayed onto or have landed near an electrode bath later replaced.

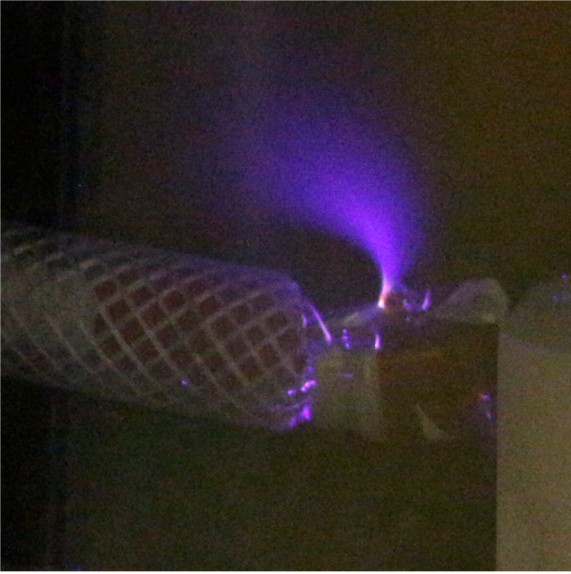

**Figure A8.** A 5-s exposure photograph of an experimental high voltage contact design which has not yet been optimized to reduce corona discharge. The source of the large corona discharge is a sharp edge of cut Kapton® tape. This high voltage contact design was later replaced by the one described in the main text.

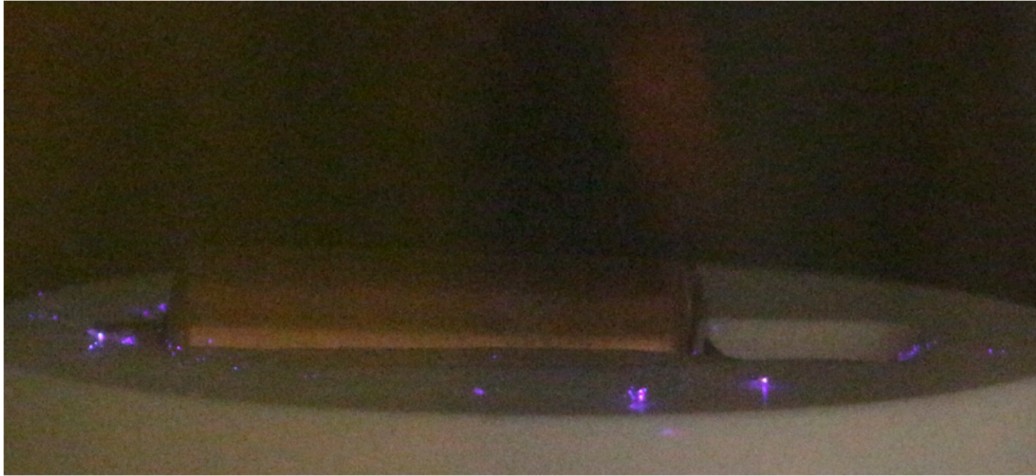

**Figure A9.** A 5-s exposure photograph of corona discharge on the solution bath near the rotating electrode. The discharge is occurring at the tips of electrospun fibers and fragments which have fallen onto the solution bath.

*Appendix A.6. Additional Electrode Designs for Taylor Cone Templating*

As described in the paper, Taylor cone templating is the initiation of electrospinning from a near Taylor cone state. In this regard, a large variety of electrodes can be conceived. From needles removed from solution as previously described [28,29], to new designs creating strands while opening two large plates (akin to quickly pulling apart two fingers with honey between them to create thin strands). In no-voltage tests, we found that designs which incorporated quick movement faster than the solution flow rate was most likely to generate many small strands instead of fewer larger strands. In practice, creating versions of these designs within our time limitations which could be safely charged to 60+ kV limited our ability to test them.

One design which we conceived, and expect to be both easy to fabricate as well as likely to be productive is the counter-rotating carding comb. The concept is the same as the carding comb discussed in the main text, but with the addition of a counter-rotating comb (Figure A10). Instead of a hand-built wire comb, the design pictured in Figure A10 should be easy to produce with a CNC mill, waterjet, or laser cutting device. Other dislike objects which pull apart, such as gears, could also be used.

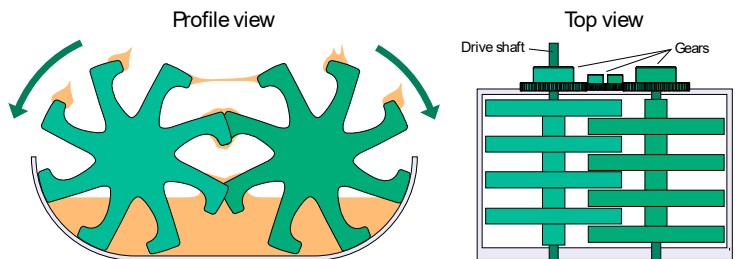

**Figure A10.** Counter-rotating carding brushes. Pinwheel like disks rotate in a bath such that they smear and then stretch strands which initiate electrospinning.

*Appendix A.7. Unsuccessful Design Concepts*

The presented design is a refined device that was developed and engineered from iterative prototypes. A few comments on unsuccessful aspects may be useful to consider in future designs. To keep costs low, the initial build incorporated wood for some parts (later replaced by StarBoard®). This was very useful for creating a mockup, but the "dry" wood and or wood glue was conductive enough that we observed high voltages across its surface. The large volume of these parts resulted

in high-voltage losses and an overall electric field that was not optimal for electrospinning. Other authors have reported successful incorporation of wooden parts [14], however, at the high voltages encountered in our work we do not recommend this. A version of the electrospinner intended to expose spun fibers to elevated temperatures (~400 °C) was also designed, incorporating cordierite ceramic parts. We learned that the quality of the ceramic glazing is especially important, crazing (a glaze defect in which a fine network of surface cracks is formed) significantly degraded the dielectric properties of our ceramic parts. Later we determined the advantage of the high temperature was its influence on relative humidity, and that the temperature could degrade to the fiber morphology. Due to these reasons, this design was not further developed.

*Appendix A.8. Material and Time Costs*

Material costs for the device were less than $3000 USD, excluding the high voltage power supplies and fume hood. It took one of the authors and a workshop assistant approximately 160 h (combined total of author + assistant) to plan construction and fabricate the reported device. Several days were also spent installing safety features and grounding parts of the fume hood, a task that was vital to the operation of the device but may vary significantly depending on the age and design of the fume hood.

**Appendix B**

*Appendix B.1. Fiber Diamater Measurments and Uncertainty Budget*

As is best practice for reporting measured values from micrographs [34], we have constructed an uncertainty budget as outlined in the "Evaluation of measurement data—Guide to the expression of uncertainty in measurement" [35]. We have modeled our uncertainty budget after Crouzier et al., which discuss the construction of an uncertainty budget for nanoparticle diameter determination and use a similar Zeiss scanning electron microscope [34]. Adobe CS3 and CC graphic software (Illustrator, Fireworks, and Photoshop) were used to prepare and analyze fibers.

Initially, we attempted to use DiameterJ, an image software intended for nanofiber measurements [36]. However, we observed undercounting of fiber diameter by the software relative to human counts, similar to the DiameterJ authors' original finding (Figure 5C and D in [36]). Our hand-produced measurements were in close agreement to a sample of more time-consuming measurements made following the work of Crouzier et al. [34]. As reported, DiameterJ was substantially faster, and collected many more datapoints. Contrary to their findings, this was at expense of our interpreted accuracy. Using the test dataset provided in [37], we were able to reproduce their results, which was especially useful for confirming that we used the software correctly. Although Crouzier et al. [34] measured nanoparticles (not fibers), their use of measurement standards is well discussed in the literature, and their well-described uncertainty budget supports our decision to use their approach for quantification of our described measurements. We do not report any fiber values obtained from DiameterJ.

Fiber diameters were measured perpendicular to the longest dimension in settings where the edges of the fiber could be clearly distinguished, typically, in sections where the background was black or high contrast edges were unambiguously distinguishable. First, edges were identified along the fiber length. Second, we located edge pairs on the right and left sides of the fiber. Finally, a measurement was taken between the two identified edges. Each fiber was only measured once at the largest diameter along sections with clear edge pairs. The work of Öznergiz et al. describe a similar application of edge finding to nanofiber analysis [38]. The edges of the fibers are diffuse in the micrographs, which adds uncertainty as to where the physical edge of the fiber begins. Crouzier et al. and Delvallée et al. both discuss this problem in relation to their nanoparticles, and determined that selecting the full width half max (FWHM) of their features (all smaller than 100 nm) resulted in measurements most accurate to AFM measurements [34,39]. However, we were concerned with the transferability of the high-quality SEM work demonstrated in both publications to our own capabilities, and instead decided

upon base measurement (described in Crouzier et al. as $D_{Eq\text{-}base}$). According to previous work, a base measurement will result in an over estimation of the diameter [34,39]. No image threshold was selected due to complication of the fiber mat presenting both blank background, and fiber on fiber backgrounds. Previous work suggests that no threshold will also result in an overestimation of the diameter [34]. Crouzier et al. recommends 3 kV [34], but our micrographs were taken with 10 kV. We do not have any quantifications at our higher voltage, but using the data from 2 to 5 kV present in Crouzier et al. [34], we estimate ± 2 nm uncertainty in our readings. Crouzier et al. [34] presents an upper limit for pixel size. In Table A1, we compare the ratio of Crouzier et al.'s [34] recommendation to the feature sizes of this work to validate our pixel sizes and our use of the pixel size uncertainty contribution from Crouzier et al.

**Table A1.** Comparison of the limit on pixel size to feature size ratio from Crouzier et al. and this work.

| Source | Feature Size (nm) | Pixel Size (nm) | Pixel to Feature Ratio |
|:---:|:---:|:---:|:---:|
| Crouzier et al. [34] | ~29 | 2.8 | 1:10 |
| Figure A12 | 120 | 1.5 | 1:80 |
| Figure A13 | 833 | 34.5 | 1:24 |
| Figure A14 | 212 | 20.4 | 1:10 |

In comparing detailed fiber determinations as pictured in Figure A11 to the previously described edge method, we determined an uncertainty in fiber edge due to image quality is ±0.87% relative to the fiber diameter. Our measurements of fiber diameters were human reproducible within ±0.24% relative to the fiber diameter. We determined that contrast and brightness adjustments on collected micrographs result in ±1.01% uncertainty relative to the fiber diameter. Our platinum coating is estimated to vary the observed thickness ±1 nm. For reasons discussed in Crouzier et al. [34], we did not account for environmental conditions such as humidity. Leading edge distortion was not considered relevant since we had abundant images and could select for objects within the central 80% of the image. The limited depth of focus at high magnification, and the vertical distribution of fibers within our fiber mat were difficult to study, as evident in Figure A12. In this sample, we performed diameter measurements on in- and out-of-focus fibers. Due to our conservative measurement technique, we expect that defocused fiber diameters were overestimated relative to the real diameter, but we did not pursue quantifying this uncertainty. Tables A2–A4 list our constructed uncertainty budgets for the samples featured in Figures A12–A14, respectively.

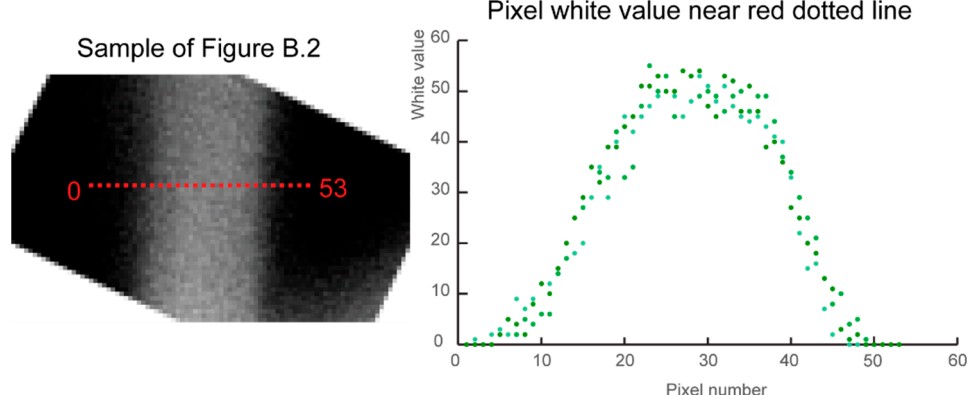

**Figure A11.** A detailed characterization of a fiber diameter (selected from Figure A12). The sample of Figure A12 depicts the micrograph of an example fiber, and the Pixel white value near red dotted line displays the white value of the pixel (a value from 0 to 100) for three transects across the fiber. In this example 1 pixel is equivalent to 1.52 nm. The baseline value would be selected near 9 and 47, resulting in a total pixel diameter of 38 pixels, or 57.8 nm.

**Table A2.** Uncertainty budget for measured values presented in relation to Figure A12.

| Uncertainty Component Description | Estimated Uncertainty (nm) | Type | Probability Distribution | Standard Uncertainty | % Contribution |
|---|---|---|---|---|---|
| Pixel size [1] | 0.014 | B | 1σ | 0.014 | 0.0% |
| Repeatability | 0.3 | A | 1σ | 0.3 | 1.1% |
| Magnificaiton [2] | 0.26 | B | 1σ | 0.26 | 0.8% |
| Beam Width [2] | 1.7 | B | 1σ | 1.7 | 35.9% |
| Operator selection [2] | 0.2 | B | 1σ | 0.2 | 0.5% |
| Operating voltage | 2 | B | Rect. | 1.1547 | 16.6% |
| Contrast-Brightness | 1.21 | A | 1σ | 1.21 | 18.2% |
| Fiber edge | 1.04 | A | 1σ | 1.04 | 13.4% |
| Human-power | 0.29 | A | 1σ | 0.29 | 1.0% |
| Platinum coating | 1 | B | 1σ | 1 | 12.4% |
| Combined Uncertainty: | | | | 2.8 | |
| k (Coverage Factor): | | | | 2.87 | |
| Expanded Uncertainty: | | | | 8.1 | |

[1] See Table A1 and the discussion in the text on pixel size to feature size ratios. [2] Adapted from the uncertainty estimation of Crouzier et al.

**Table A3.** Uncertainty budget for measured values presented in relation to Figure A13.

| Uncertainty Component Description | Estimated Uncertainty (nm) | Type | Probability Distribution | Standard Uncertainty | % Contribution |
|---|---|---|---|---|---|
| Pixel size [1] | 0.014 | B | 1σ | 0.014 | 0.0% |
| Repeatability | 0.3 | A | 1σ | 0.3 | 0.1% |
| Magnificaiton [2] | 0.26 | B | 1σ | 0.26 | 0.1% |
| Beam Width [2] | 1.7 | B | 1σ | 1.7 | 2.2% |
| Operator selection [2] | 0.2 | B | 1σ | 0.2 | 0.0% |
| Operating voltage | 2 | B | Rect. | 1.1547 | 1.0% |
| Contrast-Brightness | 8.4133 | A | 1σ | 8.4133 | 53.3% |
| Fiber edge | 7.2471 | A | 1σ | 7.2471 | 39.6% |
| Human-power | 1.9992 | A | 1σ | 1.9992 | 3.0% |
| Platinum coating | 1 | B | 1σ | 1 | 0.8% |
| Combined Uncertainty: | | | | 11.5 | |
| k (Coverage Factor): | | | | 4.5 | |
| Expanded Uncertainty: | | | | 52.2 | |

[1] See Table A1 and the discussion in the text on pixel size to feature size ratios. [2] Adapted from the uncertainty estimation of Crouzier et al.

**Table A4.** Uncertainty budget for measured values presented in relation to Figure A14.

| Uncertainty Component Description | Estimated Uncertainty (nm) | Type | Probability Distribution | Standard Uncertainty | % Contribution |
|---|---|---|---|---|---|
| Pixel size [1] | 0.014 | B | 1σ | 0.014 | 0.0% |
| Repeatability | 0.3 | A | 1σ | 0.3 | 0.7% |
| Magnificaiton [2] | 0.26 | B | 1σ | 0.26 | 0.5% |
| Beam Width [2] | 1.7 | B | 1σ | 1.7 | 21.1% |
| Operator selection [2] | 0.2 | B | 1σ | 0.2 | 0.3% |
| Operating voltage | 2 | B | Rect. | 1.1547 | 9.8% |
| Contrast-Brightness | 2.1412 | A | 1σ | 2.1412 | 33.5% |
| Fiber edge | 1.8444 | A | 1σ | 1.8444 | 24.9% |
| Human-power | 0.5088 | A | 1σ | 0.5088 | 1.9% |
| Platinum coating | 1 | B | 1σ | 1 | 7.3% |
| Combined Uncertainty: | | | | 3.7 | |
| k (Coverage Factor): | | | | 2.87 | |
| Expanded Uncertainty: | | | | 10.6 | |

[1] See Table A1 and the discussion in the text on pixel size to feature size ratios. [2] Adapted from the uncertainty estimation of Crouzier et al.

*Appendix B.2. Demonstration on Select Compositions*

The yield and diameter of fibers were sensitive to changes in system parameters, especially solution viscosities and relative humidity. We characterized fiber yields of 0.5 to 2 g per hour using a solution of 3.8 wt% polyvinylpyrrolidone, 4.3 wt% $CsH_2PO_4$, 63.4 wt% water, and 28.5 wt% ethanol, a cylindrical electrode 3 inches long with a 1.25 inch diameter rotating at 5 rpm, a 2-inch diameter collector drum rotating at 33 Hz (~5 m/s), 40%–48% relative humidity, air temperature of 20 °C, spinning electrode biased to +40 kV, and collector drum biased to −30 kV. We spent no effort in optimizing device, solution or environment parameters for a specific fiber morphology or size.

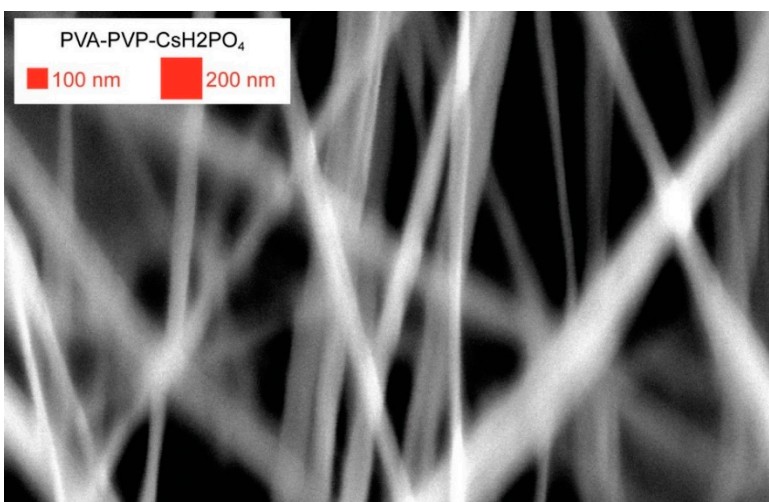

**Figure A12.** Polyvinyl alcohol- polyvinylpyrrolidone fiber containing conductive $CsH_2PO_4$. The fibers have varying diameters along their length, which we interpret to indicate a ribbon like morphology. The median diameter of this sample was 120 nm. The vertical distribution of fibers in the 3D fiber mat made image collection somewhat difficult at this magnification.

We produced the smallest fibers (Figure A12) from a solution with low viscosity and similar to compositions which have been studied previously [40]. This sample was produced from a solution of 1.2 wt% polyvinyl alcohol, 1.2 wt% polyvinylpyrrolidone, 5.3 wt% $CsH_2PO_4$, 46.4 wt% water, and 45.9 wt% ethanol, spun on a cylinder electrode (−30 kV) in 40% relative humidity, 13.5 cm below the

collector (+45 kV) with a total voltage drop of 75 kV. The surface speed of the collector was 14.9 m/s and a yield of 1.4 g/hr was collected. Microscopy of collected samples indicated minimum diameter of 39 nm, maximum diameter of 403 nm. In 116 analyzed fibers, median diameter was 120 nm, mean diameter was 132 nm, first quartile 90 nm and third quartile at 160 nm. The uncertainty of individual measurement was estimated to ±8.1 nm, as presented in Table A1 and discussed in Appendix B.1. Dynamic viscosity at bath conditions was 3 poise.

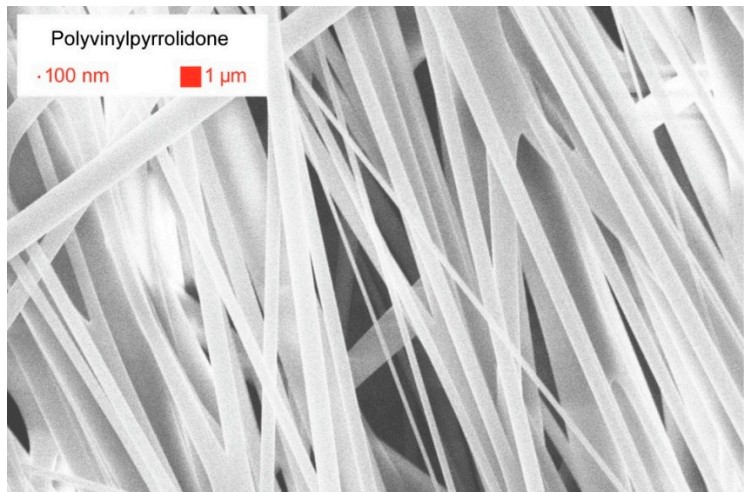

**Figure A13.** Polyvinylpyrrolidone fibers produced from our device. Median diameter of fibers from this sample was 833 nm.

Figure A13 was produced from a solution of 12 wt% polyvinylpyrrolidone and 88 wt% methanol, spun on a cylinder electrode (−20 kV) in 33% relative humidity, 12 cm below the collector (+40 kV) with a total voltage drop of 60 kV. The surface speed of the collector was 24.4 m/s. Microscopy of collected samples indicated minimum diameter of 202 nm, maximum diameter of 2.67 μm, with occasional features 3 to 10.8 μm large most likely the result of undried polymer flung onto the collector. In 100 analyzed fibers, median diameter was 833 nm, mean diameter was 881 nm, first quartile 584 nm and third quartile at 1.07 μm. The uncertainty of individual measurement was estimated to ±52.2 nm, as presented in Table A1 and discussed in Appendix B.1.

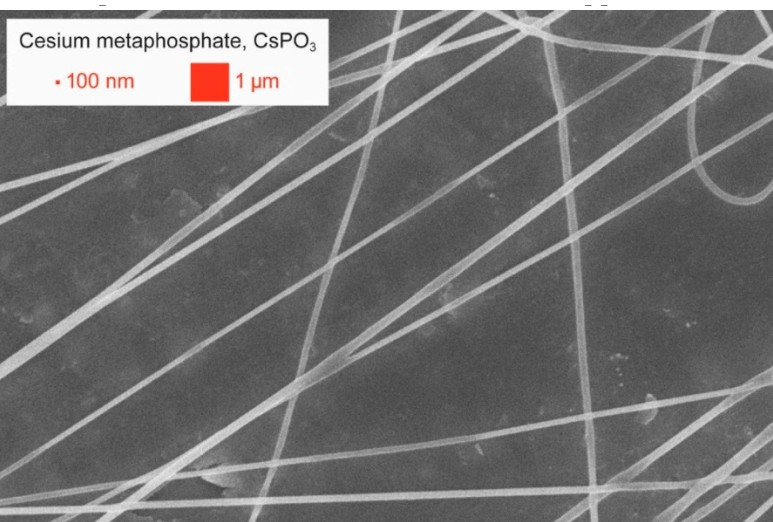

**Figure A14.** Cesium metaphosphate electrospun fibers produced from our device. Median diameter of fibers from this sample was 212 nm.

Figure A14 was produced from a solution of 69 wt% CsPO3, 31 wt% water which then allowed to evaporate. The viscosity was observed to increase as the concentration of the solvent (water) decreased relative to the solute (CsPO3). Addition of alcohol would induce immediate precipitation of the solute. Addition of polyvinylpyrrolidone or polyvinyl alcohol to high-concentration metaphosphate solutions would induce immediate coagulation of a white rubber-like solid. Electrospinning of a similar solution from a capillary needle has been previously described [41], but this is the first larger volume need-free demonstration of this solution. Reports of electrospinning other metaphosphates employed a carrier polymer, which is not used in our test [42,43]. It was spun on a saguaro electrode (−20 kV) in 36% relative humidity, 12 cm below the collector (+50 kV) with a total voltage drop of 70 kV. The surface speed of the collector was 4.9 m/s. Microscopy of collected samples indicated minimum diameter of 86 nm, maximum diameter of 1.01 μm. In 94 analyzed fibers, median diameter was 212 nm, mean diameter was 245 nm, first quartile 174 nm and third quartile at 282 nm. The uncertainty of individual measurement was estimated to ±10.6 nm, as presented in Table A1 and discussed in Appendix B.1. Dynamic viscosity at bath conditions was 110 poise, the solution exhibits shear thinning, but this was not further investigated.

*Appendix B.3. System Paramaters*

Parameters relevant to electrospinning include device parameter (electric field, electrode-collector distance, rotating collector speed, etc.), solution parameters (surface tension, conductivity, viscosity, solution temperature, etc.), and environment parameters (air temperature, relative humidity, air pressure, etc.). Our device was intended to present a large range of device parameters such that operators were unlikely to be limited by device conditions when preforming experimental tests. Table A5 reports the range of parameters that are influenced by the design, materials and construction of this particular device. Table A6 reports environmental parameters that we achieved by incorporating accessory equipment inside the fume hood.

**Table A5.** Range of possible system parameters capable of being studied with this device.

| Parameter | Minimum | Maximum | Units |
|---|---|---|---|
| Total bias | 0 | 93,000 [1] | volts |
| Collector electrode distance at peak voltage | 11 | 33 | cm |
| Electrode rotational frequency | 0.03 | 0.2 | Hz |
| Electrode rotational frequency | 2 | 12 | rpm |
| Collector rotational frequency | 0 | 105 | Hz |
| Collector rotational frequency | 0 | 6300 | rpm |
| Collector surface speed | 0 | 50 | m/s |

[1] Limited by power supply, not materials or device performance.

**Table A6.** Environment parameters controlled by accessory equipment.

| Parameter | Minimum | Maximum | Units | Source |
|---|---|---|---|---|
| Air temperature | 19 | 55 | C | External air heaters |
| Air flow | 0 [1] | 0.6 | m/s | Fans and fume hood |
| Relative humidity | 15 [2] | 85 | % | Dehumidifier or humidifiers |

[1] Airflow was below the limit of detection. [2] Lower relative humidity was achievable on dry days, and would also be achievable with better dehumidification equipment.

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
