# Peer review of "An Adaptable Device for Scalable Electrospinning of Low- and High-Viscosity Solutions"

_instruments, doi:10.3390/instruments3030037_

Round 1

Reviewer 1 Report

The paper failed to establish the applicability of the instrument in actual electrospinning. Despite rigorous efforts of authors this is an incomplete work, because-

in contrast to authors' claims in abstract there is no discussion on applicability of the said instrument in conclusion section

there is no image of electrospun nanofiber, nanofiber mat, SEM images

there is no process parameter study.

Reviewer 2 Report

Authors came up with modified electrospinning set up and claimed this set up can be used to produced nanofibers. But the polymer solutions tested in these experiments are not reported ex concentration, viscosity etc. 

Further, the fibers produced using this system was not represented therefore authors can not claim this system can be used to produce nanofibers. SEM images of the produced fibers have to present to claim this point. 

Therefore authors have to restructure this paper including missing data

Round 2

Reviewer 1 Report

Few things that bothered me in the revision file-

There are no references beyond 21 in main reference section, whereas references beyond 21 are mentioned in the main manuscript, which is kind of sloppy. Figure B1, is extremely poor and completely hazy to understand anything from them. This baffles me that how did authors measure fiber sizes from them. Same is with Figure B2. I would like to understand how a metal salt can be spun which clearly has no viscoelastic property. Since this manuscript refers an article, which is not mentioned in the reference section, it is bit controversial. A similar article from Journal of the Korean Ceramic Society Vol. 44, No. 5, pp. 244~247, 2007, if that's what authors are trying to refer who has shown a case of Electrospun Calcium Metaphosphate Nanofibers, then authors should understand that a carrier polymer like PVP was necessary to form fibrous architecture. 

Reviewer 2 Report

I accept the present form of the manuscript 

Author Response

Dear Reviewer 2,

In response to comments by Review 1, we have added additional references in the introduction. Hopefully this improves “Does the introduction provide sufficient background and include all relevant references” for you as well.

Once again, thank you for your time and review of our work!

Sincerely,

-Ryan J McCarty

Round 3

Reviewer 1 Report

I understand the authors are trying to build a low cost electrospinning unit which has its large set of advantages and the efforts put in by the authors are highly appreciated. However, authors should also understand the fact that the beauty of the method will only be appreciated by others when tangible nanofibers are produced and the images should support them. Authors' inability to resolve the nanofibers or the operator's fault in setting a proper brightness and/or contrast to elucidate the fibrous architecture is minor glitch in producing good result. I am still quite uncertain about the third result of metal salt spinning without a polymer carrier. I would like to advice authors to provide some supplementary videos to support their claims. 

Author Response

Thank you for your persistence and additional requests and concerns. We have responded to point 1, and have created a supplemental video to meet Point 2. The detail responses can be seen in the attachment.
